# Optimizing the Schedule of PARP Inhibitors in Combination with ^177^Lu-DOTATATE: A Dosimetry Rationale

**DOI:** 10.3390/biomedicines9111570

**Published:** 2021-10-29

**Authors:** Andreas Hallqvist, Johanna Svensson, Linn Hagmarker, Ida Marin, Tobias Rydén, Jean-Mathieu Beauregard, Peter Bernhardt

**Affiliations:** 1Department of Oncology, Institute of Clinical Science, Sahlgrenska Academy, University of Gothenburg, 41345 Gothenburg, Sweden; johanna.b.svensson@vgregion.se; 2Medical Physics and Biomedical Engineering, Sahlgrenska University Hospital, 43415 Gothenburg, Sweden; linn.hagmarker@gu.se (L.H.); peter.bernhardt@gu.se (P.B.); 3Department of Medical Radiation Sciences, Institute of Clinical Science, Sahlgrenska Academy, University of Gothenburg, 43415 Gothenburg, Sweden; ida.marin@vgregion.se (I.M.); tobias.ryden@phonsa.se (T.R.); 4Department of Medical Imaging, CHU de Québec-Université Laval, Quebec City, QC G1R 3S1, Canada; jean-mathieu.beauregard@crchudequebec.ulaval.ca; 5Department of Radiology and Nuclear Medicine, Université Laval, Quebec City, QC G1R 3S1, Canada

**Keywords:** ^177^Lu-DOTATATE, PARP inhibitor, somatostatin positive tumor, olaparib

## Abstract

^177^Lu-DOTATATE for neuroendocrine tumours is considered a low-toxicity treatment and may therefore be combined with other pharmaceuticals to potentiate its efficacy. One approach is to add a poly-[ADP-ribose]-polymerase (PARP) inhibitor to decrease the ability of tumour cells to repair ^177^Lu-induced DNA damage. To decrease the risk of side effects, the sequencing should be optimized according to the tumour-to-normal tissue enhanced dose ratio (*TNED*). The aim of this study was to investigate how to enhance ^177^Lu-DOTATATE by optimal timing of the addition of a PARP inhibitor. Biokinetic modelling was performed based on the absorbed dose to the bone marrow, kidneys and tumour; determined from SPECT/CT and planar images from 17 patients treated with ^177^Lu-DOTATATE. To investigate the theoretical enhanced biological effect of a PARP inhibitor during ^177^Lu-DOTATATE treatment, the concept of relative biological effectiveness (*RBE*) was used, and PARP inhibitor administration was simulated over different time intervals. The absorbed dose rate for the tumour tissue demonstrated an initial increase phase until 12 h after infusion followed by a slow decrease. In contrast, the bone marrow showed a rapid initial dose rate decrease. Twenty-eight days after infusion of ^177^Lu-DOTATATE, the full absorbed dose to the bone marrow and kidney was reached. Using an *RBE* value of 2 for both the tumour and normal tissues, the *TNED* was increased compared to ^177^Lu-DOTATATE alone. According to the modelling, the PARP inhibitor should be introduced approximately 24 h after the start of ^177^Lu-DOTATATE treatment and be continued for up to four weeks to optimize the *TNED*. Based on these results, a phase I trial assessing the combination of olaparib and ^177^Lu-DOTATATE in somatostatin receptor-positive tumours was launched in 2020 (NCT04375267).

## 1. Introduction

Peptide receptor radionuclide therapy (PRRT) with ^177^Lu-DOTATATE is a well-established method in the treatment of somatostatin receptor (SSTR)-positive neuroendocrine tumours and is the only SSTR-directed radiopharmaceutical formerly approved by regulatory authorities. PRRT with ^177^Lu-DOTATATE is considered a low-toxicity treatment but sometimes with insufficient efficacy, thus motivating treatment intensification with, e.g., combination therapies [1]. A theoretically interesting approach is to combine ^177^Lu-DOTATATE with a poly-[ADP-ribose]-polymerase (PARP) inhibitor such as olaparib to decrease the ability of tumour cells to repair DNA damage. The addition of a PARP inhibitor might overcome the repair-dependent survival of tumour cells and increase the probability of tumour cell death [2]. ^177^Lu decays with the greater part of the emitted energy as β-particles, i.e., electrons, and mainly causes DNA single strand breaks (SSBs). However, the cell killing effect of ionizing radiation in most cases is due to double strand breaks (DSBs), as DSBs are much more difficult to repair. Consequently, there must be a high number of SSB-damaging hits to a cell to induce cell death [2]. PARPs are enzymes that become activated by DNA damage where they bind to areas of SSBs and recruit other proteins, initiating a repair complex. During this process, PARP is released from the damaged site and the damage may be accessed by the repair proteins. The binding of a PARP inhibitor results in PARP becoming trapped on the DNA which results in a DNA lesion that cannot be bypassed by the replication fork. This leads to the formation of DSBs and stalled replication forks at the site of damage and subsequently increases the incidence of catastrophic genomic instability, which may result in cell death [3]. The addition of a PARP inhibitor that specifically targets the repair mechanism to ionizing radiation that causes SSBs will make the cells more sensitive to β-emission [1]. In the preclinical work-up for such a combination strategy, it is important to consider the possible additional toxicity and assess the best way of sequencing the therapies to optimize the tumour-to-normal tissue enhanced dose ratios (*TNED*s), i.e., maximize the tumoricidal synergistic effect while striving to minimize the same effect to the organs at risk.

The aim of the current work was to find the best possible theoretical time window for the addition of olaparib relative to the ^177^Lu-DOTATATE administration through a modelling study as a base for future clinical testing. We hypothesized that the biokinetics of the tumour and normal tissues differ, and this difference may be used to find suitable time points for the start and finish of the administration of olaparib. Furthermore, we investigated if the addition of olaparib could have the potential to increase the therapeutic window, i.e., that the effect on tumour response would be larger than that on normal tissues, with an optimal schedule for olaparib administration. Based on the present results, we have designed and launched a clinical prospective phase I trial to assess the safety of the combination of ^177^Lu-DOTATATE with escalated doses of olaparib in patients with SSTR-positive tumours (NCT04375267).

## 2. Materials and Methods

Seventeen patients were selected from a cohort of 51 patients treated with ^177^Lu-DOTATATE up to an absorbed kidney dose of 27 Gy during 2006–2011, and the absorbed dose rate in tumours, bone marrow and kidneys was studied. The retrospective use of the patient data and waiver of consent were approved by the Regional Ethical Review Board in Gothenburg.

The renal absorbed doses were estimated from planar images as previously described [4]. The absorbed dose to the bone marrow was estimated by a constructed two-compart imaging model [5]. The present study also requires absorbed dose to tumours. A prerequisite for accurate estimation of the activity and size of tumours is the use of both SPECT/CT as well as repeated planar images after treatment. To obtain a detailed description of the biokinetics after infusion, four planar images were collected at 2, 24, 48 and 168 h.p.i. The tumour uptake in the planar images also had to be easily visualized.

### 2.1. Gamma Camera Acquisition

The gamma cameras used for the examinations during this period were Millennium VG Hawkeye with a crystal thickness of 5/8″ and Infinia Hawkeye 4 (General Electric Medical Systems, Milwaukee, WI, USA) with a crystal thickness of 3/8″, both equipped with a medium-energy parallel-hole collimator. Planar whole-body scans were performed with a scanning time of 10 min and a 20% energy window over the 208 keV photon peak. The clinical SPECT images were acquired 24 h.p.i. 7.4 GBq ^177^Lu-DOTATATE, with a 30-s frame time duration for 120 projections. The matrix size was 128 × 128 with a pixel size of 4.42 mm and a slice thickness of 4.42 mm. The CT images used in the SPECT/CT reconstructions were acquired with a Millennium VG Hawkeye system.

For SPECT/CT reconstruction, we adopted a Monte Carlo-based reconstruction technique by using the Sahlgrenska Academy Reconstruction code (SARec) [6]. In the code, the forward projections included simulation of photon attenuation, scattering, and collimator resolution with septal penetration. The scattering in the collimator is approximated with an experimentally determined photon-scattering kernel. The SPECT/CT reconstructions use 6 subsets and 10 iterations, with a simulation time of three minutes.

### 2.2. Dosimetry

The absorbed doses to the kidney and bone marrow of this patient cohort have been presented previously [4,5,7,8], and the dosimetry methods for these organs are briefly explained below. Tumour absorbed doses were assessed by a hybrid imaging method, i.e., the ^177^Lu-DOTATATE kinetics were determined in the planar images at time points 2, 24, 48 and 168 h.p.i., and the activity concentration was determined in an SPECT/CT image collected 24 h.p.i.

The most easily visualized tumour in the planar image within the field of the SPECT image was selected for further dosimetry evaluation and was identified on diagnostic CT images. The volume of interest (VOI) on the SPECT/CT and the region of interest (ROI) on the planar images were manually drawn for the selected tumour (Figure 1). On the planar images, a background ROI was also created. These ROIs were created on the planar image at 24 h.p.i. and used on the other three planar images. The activity in the tumour ROI was subtracted by the activity in the background ROI and adjusted for the difference in the ROI size.

The activity concentrations in the SPECT VOIs were corrected for partial volume effects by a recovery coefficient *RC*. The *RC* was determined from measurements of six spheres with known activity concentrations in a Jaszczak phantom. The inner diameters of the spheres were 10, 12, 16, 20, 25 and 31 mm. The *RC* for each sphere was determined by dividing the measured activity concentration by the known activity concentration. A sigmoid function was fitted to the *RC* in MATLAB (Mathworks, CA, USA):$$RC\left(d\right)=\frac{1}{1+{\left(\frac{a}{d}\right)}^{b}}$$
where *d* is the inner sphere diameter, and *a* and *b* are fit coefficients.

The activity concentration in the planar images was determined by:$$C{\left(t\right)}_{Tu}=C_{SPECT,\;24h}\cdot \frac{A{\left(t\right)}_{Tu}}{A{\left(24\right)}_{Tu}}\cdot RC\left(d\right)$$

The tumour dose rate was determined by:$$\dot{D}{\left(t\right)}_{Tu}=C{\left(t\right)}_{Tu}\cdot \emptyset_{e}\cdot E_{e}\cdot k$$
where *C*(*t*)*_Tu_* is in units of MBq/g, and *k* converts the dose rate to units of Gy/h. For all tumours, it was assumed that the energy of emitted electrons (*E_e_* = 147 keV/decay) was locally absorbed, i.e., the absorbed fraction ($\emptyset_{e})$ was equal unity. Any absorption of photons was neglected.

For kidney dosimetry, we used only the left kidney since it overlapped less with surrounding high uptake areas on the planar image. The activity concentration, *C_k_*, was determined in the work by Svensson et al. by the conjugate view method [4]. The kidney mass was estimated from CT images. By assuming that the energy of all emitted electrons and the absorbed fraction ($\emptyset_{e})$ is equal unity and neglecting any absorption of photons, the absorbed dose rate (Gy/h) to the kidneys was determined by:$$\dot{D}{\left(t\right)}_{Ki}=C{\left(t\right)}_{Ki}\cdot \emptyset_{e}\cdot E_{e}\cdot k$$

The absorbed dose rate to the bone marrow was determined by the two-compartment imaging method, as previously described [5,7], and later refined by Hagmarker et al. [8] by adopting a calibration factor (*CF*) based on individual measurements. The *CF* used in this work was set to 3.2, as proposed by Hagmarker et al. The estimate of the absorbed dose rate to bone marrow with the two-compart model is:$$\dot{D}{\left(t\right)}_{BM}=A{\left(t\right)}_{low}\cdot m_{low}^{-1}\cdot \emptyset_{e}\cdot E_{e}\cdot CF\cdot k+A{\left(t\right)}_{low}\cdot S_{BM\leftarrow low}+A{\left(t\right)}_{high}\cdot S_{BM\leftarrow high}$$
where $A{\left(t\right)}_{low}$ and $A{\left(t\right)}_{low}$ are the activity in the low uptake and high uptake areas in the planar image, respectively. $m_{low}^{-1}$ is the estimated mass of the low uptake area. $S_{BM\leftarrow low}$ and $S_{BM\leftarrow low}$ are the *S*-values for low- and high-uptake area irradiation of the BM, respectively. A more detailed description of the model can be found in our previous studies [5,7,8].

A challenge for estimation of the absorbed dose rate during therapy is to accurately estimate the pharmacokinetics of the radiopharmaceutical in the tissue of interest. In the present study we used four data points. Three of them were within the first two days, making it possible to capture the initial dose rate differences between the tumour and the normal organs at risk: kidney and bone marrow. Biexponential fits were used to describe the dose rate during therapy:$$\dot{D}{\left(t\right)}_{Tissue}=C_{1}\cdot e^{at}+C_{2}\cdot e^{bt}$$
where *t* is time p.i. ^177^Lu-DOTATATE and *C*_1_, *C*_2_, *a*, *b* are the fitting parameters. The curve fit was performed with MATLAB 2019b.

In radiotherapy, the goal is to find an optimal therapeutic time window where the absorbed dose to the tumour relative to the critical normal tissue is as high as possible. From the dose rates above, the mean absorbed dose between different time points, p.i., to a tissue can be estimated by:$$D_{Tissue}=\int_{0}^{t_{1}}\dot{D}{\left(t\right)}_{Tissue}dt+\int_{t_{1}}^{t_{2}}\dot{D}{\left(t\right)}_{Tissue}dt+\int_{t_{2}}^{\infty }\dot{D}{\left(t\right)}_{Tissue}dt$$

The tumour-to-normal tissue absorbed dose ratio (*TND*) within different time intervals is estimated by:$$TND_{NT,t_{i}-t_{j}}=\frac{D_{Tu,t_{i}-t_{j}}}{D_{NT,t_{i}-t_{j}}}$$
where *NT* indicates the normal tissue of interest and *t_i_ − t_j_* is the time interval. In this study, we specifically investigated the following time intervals: 0–∞, 0–6 h, 6 h–28 d, 6 h–∞, 24 h–28 d, 24 h–∞ and 28 d–∞.

To investigate the theoretical enhanced biological effect of adding a cytotoxic pharmaceutical during ^177^Lu-DOTATATE, we used the concept of relative biological effectiveness (*RBE*) and inserted it into the time interval when the cytotoxic pharmaceutical was administered. The estimated enhancement absorbed dose (*ED*) for a tissue is then:$$ED_{Tissue,RBE}=\int_{0}^{t_{1}}\dot{D}{\left(t\right)}_{Tissue}dt+RBE\cdot \int_{t_{1}}^{t_{2}}\dot{D}{\left(t\right)}_{Tissue}dt+\int_{t_{2}}^{\infty }\dot{D}{\left(t\right)}_{Tissue}dt$$

The tumour-to-normal tissue enhancement dose ratio (*TNED*) within different time intervals is estimated by:$$TNED_{RBE_{Tu}/RBE_{NT}}=\frac{ED_{Tu,RBE_{Tu}}}{ED_{NT,RBE_{NT}}}$$

In a preclinical study, the addition of a PARP inhibitor to the treatment with ^177^Lu-DOTATATE caused an increased additional effect in cultured cells [2]. The relative biological effect (*RBE*) derived from this study can be estimated to approximately 2. By using this *RBE* value for tumor cells and varying *RBE* between 1 and 2 for the normal tissue, the *TND* and *TNED* were estimated.

## 3. Results

The absorbed dose rates of ^177^Lu-DOTATATE in tumours, bone marrow and kidneys are shown in Figure 2. The biexponential models had an excellent fit to the data points (r > 0.98). The dose rate for the tumour tissue demonstrated an initial increase phase until approximately 12 h after injection. Thereafter the dose rate slowly decreased (Figure 2a). In contrast, the bone marrow showed a rapid initial dose rate decrease, without a measurable initial uptake phase, which after approximately 10–24 h was shifted to a slower decreasing dose rate (Figure 2b). The kidney dose rate did not possess a similar rapid initial dose rate change (Figure 2c). However, a biexponential model had to be used to capture an appropriate fit to the data points.

From the absorbed dose rate curves, the absorbed doses were estimated by integrating the area under the curve to infinity. The median absorbed doses were 0.30, 4.5 and 27.3 Gy, respectively, for the bone marrow, the kidney and the tumour (Table 1). At 28 days after the injection of ^177^Lu-DOTATATE, the absorbed doses to these tissues reached 99.9, 99.8 and 98.6% of the total dose (Figure 3). Figure 4 demonstrates the change in median TND over time for the normal tissues in the bone marrow and kidney. TNDs increased rapidly in the first 6 h; however, the TND for bone marrow still rapidly increased until approximately 24 h after administration of ^177^Lu-DOTATATE.

The median TNDs for bone marrow and kidney were estimated to be 67 and 6.1, respectively (Table 2). Assuming a delayed administration of olaparib by 6 or 24 h, the TND for bone marrow was increased by 22 and 40%, respectively. Interrupting olaparib treatment four weeks after the start had a minor effect on the TND compared to continuous treatment (Table 2). Similar results were observed for the TND of the kidney. Figure 5 shows the relative TNDs, where all except one patient had increased TNDs within the time frame 24 h to 28 d compared to the time frame 0 to infinity.

When administrating the PARP inhibitor within a time interval of 24 h–28 d, the *TNED* was increased in relation to the assumed *RBE* value (Table 3). Using an *RBE* value equal to 2 for the tumour and normal tissue, a small incremental benefit in *TNED* was generated compared to the TND obtained for ^177^Lu-DOTATATE-only treatment; *TNED*_2/2_ was increased by 14% for bone marrow and 5% for kidney (Table 4). Assuming a lower *RBE* for normal tissues compared to the tumour tissue, the *TNED* was further increased. For *RBE* equal to 2 for tumour and 1.5 for normal tissues, the median *TNED*_2/1_._5_ was 42% and 34% for the bone marrow and kidney, respectively. If no additional effect of the PARP inhibitor on normal tissues is assumed, *TNED*_2/1_ will allow a maximal additional increase of 85% to the bone marrow and kidney.

## 4. Discussion

We hypothesize that the tumoricidal effect of ^177^Lu-DOTATATE could be increased by adding olaparib to decrease the ability of tumour cells to repair DNA damage, thereby making them more sensitive to radiation and increasing the probability of tumour cell death. To optimize the sequencing, we performed this biokinetic modelling based on the uptake of ^177^Lu-DOTATATE in bone marrow, kidneys and tumour tissue during repeated measurements and calculated the best possible theoretical concurrent approach. The results suggest that olaparib administration should start approximately 24 h after infusion of ^177^Lu-DOTATATE to facilitate normal tissue repair, especially in bone marrow cells, and thereby decrease the risk of haematological and renal toxicity. The DNA repair mechanisms operate fast, within hours [9], and, due to the rapidly decreasing dose rate in the bone marrow, most of the DNA damage will have time to recover during 24 h. The effect in the tumour tissue hardly seems affected by such a delay; in contrast, it will increase the *TNED* and would potentially allow olaparib dose escalation. After four weeks of olaparib administration, the additional effect is negligible, and the drug could be paused. This kind of biokinetic modelling is not commonly performed, but the method has previously been used to calculate TND for various radionuclides [10].

The measured biokinetics for tumours and kidneys in this study are in agreement with published data, i.e., tumour tissue reaches its highest activity concentration after approximately 12 h, while the kidneys lack a pronounced peak and are often described with a single exponential function. In contrast, an accurate biokinetic profile regarding the bone marrow is more challenging to obtain due to the low uptake and the high risk of spill in from surrounding high-uptake areas, making activity quantification challenging [7]. Therefore, a common method to estimate bone marrow’s absorbed dose is to assume that the biokinetic profile follows that of blood. Nonetheless, this dosimetry method has not been able to demonstrate a correlation to bone marrow toxicity in larger cohort studies [11]. However, with our two-compartment method for estimating the bone marrow biokinetic profile, a dose response correlation has been demonstrated in one retrospective study and one prospective study [5,8]. We believe that this bone marrow biokinetic profile is more accurate than blood-based biokinetic profiles.

Treatment with ^177^Lu-DOTATATE is the most common PRRT radiopharmaceutical used in patients with metastatic somatostatin receptor-positive neuroendocrine tumours today due to its less toxic qualities compared to yttrium-based counterparts [12,13], and to the regulatory approval of the now established regimen of four cycles of 7.4 GBq based in the NETTER-1 trial [14]. Despite the proven beneficial effect on progression-free survival and overall survival at the population level, some patients show a lack of response or a very short response, and eventually all patients will progress. As the treatment with ^177^Lu-DOTATATE is considered rather nontoxic in comparison with other antitumoral strategies, a slight increase in toxicity may be acceptable if a significant increase in efficacy is shown. Theoretically, this could be accomplished through various strategies, such as shortening time intervals between cycles or increasing the number of cycles, increasing activity through dosimetry-based personalization or combining radiation with synergistic pharmaceuticals [15]. The latter has been studied with chemotherapy, including the addition of capecitabine, capecitabine and temozolomide and the mTOR inhibitor everolimus [16,17,18,19,20]. As the sometimes insufficient efficacy of ^177^Lu-DOTATATE may, at least in part, be explained by the predominance of SSBs in response to radiation from ^177^Lu, we think that the combination strategy with a PARP inhibitor, decreasing the ability of cells to repair DNA damage, is one of the most attractive options.

Preclinical investigations by Nonnekens et al. [2] show that cells expressing somatostatin receptors could be synergistically sensitized to PRRT by adding olaparib, as they observed increased cell death and reduced cellular proliferation compared to PRRT alone. Enhanced cell death was caused by increased number of DSBs, leading to genome instability. Others have found similar results when combining radionuclide therapy with a PARP inhibitor, e.g., olaparib and another radiopharmaceutical—^131^I, [21] and ^177^Lu PRRT with an alternative PARP inhibitor [22]. An increased effect of ^177^Lu-DOTATATE through indirect inhibition of PARP by the NAMPT inhibitor GMX1778 has also been observed [23].

When translating the combination strategy of ^177^Lu PRRT and PARP inhibitor into clinical research, it is essential to consider the risk of increased toxicity. The main dose-limiting effect in PRRT is bone marrow suppression and secondly decreased kidney function. Olaparib per se gives rise to a higher frequency of low-grade anaemia, neutropenia and thrombocytopenia, as well as low-grade diarrhoea, decreased appetite and nausea [24]. The latter can also be seen with PRRT, however transient. To decrease the risk of synergistic toxicity, especially regarding to the bone marrow, it is of outmost importance to assess the sequencing of the concurrent approach. Here, we show that the *TNED* will increase by delaying PARP inhibition until 24 h after the administration of ^177^Lu-DOTATATE, i.e., that the anti-tumour efficacy is expected to increase more than the risk of bone marrow toxicity. Based on these results, we designed a clinical phase I combination trial of olaparib and ^177^Lu-DOTATATE in patients with somatostatin receptor-positive tumours that was launched in 2020 (NCT04375267, EudraCT 2019-001700-37). It consists of four cycles of ^177^Lu-DOTATATE of 7.4 MBq administered every 8–12 weeks combined with an individually escalated dose of p.o. olaparib per cycle with the primary endpoint of safety measured as per CTCAE v 5.0. Olaparib administration starts 24 h after the infusion of ^177^Lu-DOTATATE and continues for four weeks at each cycle. Toxicity and other clinical data are followed closely and will, in combination with organ and tumour dosimetry, be used to assess the outcome as well as the underlying modelling.

## 5. Conclusions

The optimal therapeutic time window for PARP inhibition during PRRT should be set to avoid the initial low absorbed dose ratios between the tumour and the critical organs (BM in particular). Therefore, the PARP inhibitor should be introduced approximately 24 h after the ^177^Lu-DOTATATE infusion. The administration of the PARP inhibitor should continue up to four weeks; thereafter, the irradiation of tumour tissue is minor, and no significant additional synergistic effect of PARP is expected.

## Figures and Tables

**Figure 1 biomedicines-09-01570-f001:**
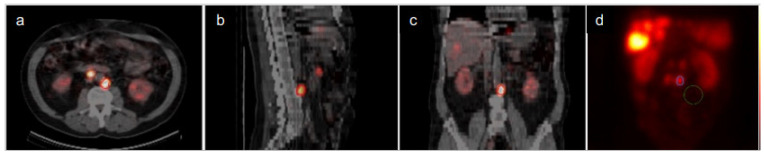
Manually drawn VOI on SPECT/CT in (**a**) transversal view (**b**) sagittal view and (**c**) coronal view. (**d**) Tumour ROI and background ROI on the corresponding planar geometrical mean image.

**Figure 2 biomedicines-09-01570-f002:**
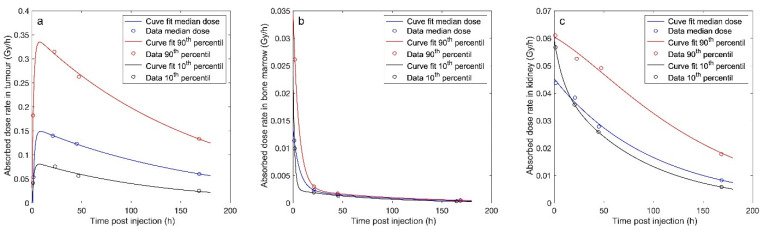
The absorbed dose rate of ^177^Lu-DOTATATE versus time after administration (post injection, p.i.) for: (**a**) tumour tissue, (**b**) bone marrow, and (**c**) kidney.

**Figure 3 biomedicines-09-01570-f003:**
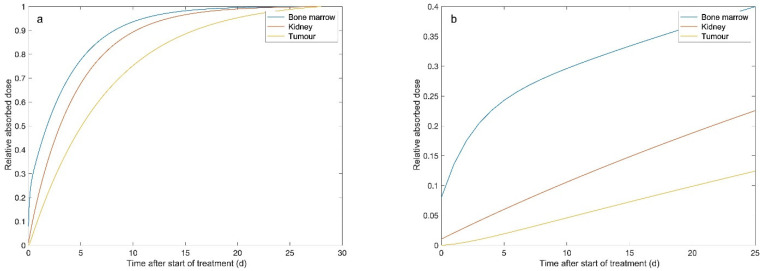
The median relative absorbed dose versus the time after treatment start with ^177^Lu-DOTATATE for the bone marrow (blue line), kidney (red line), and tumour (yellow line). Panel (**a**) shows the first 4 weeks after the start of treatment, while panel (**b**) focuses on the first 25 h after the start of treatment.

**Figure 4 biomedicines-09-01570-f004:**
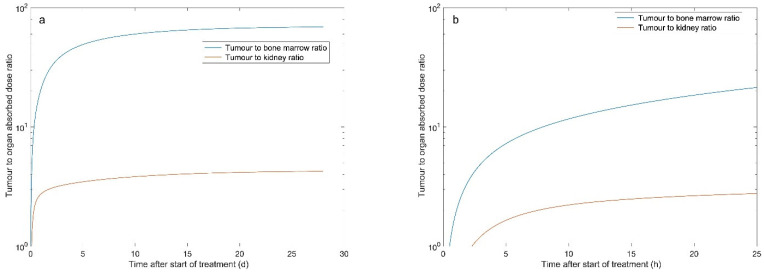
The median tumour-to-normal tissue absorbed dose ratio (TND) for the bone marrow and kidney during ^177^Lu-DOTATATE treatment. Panel (**a**) shows the first 4 weeks after the start of treatment, while panel (**b**) focuses on the first 25 h after the start of treatment.

**Figure 5 biomedicines-09-01570-f005:**
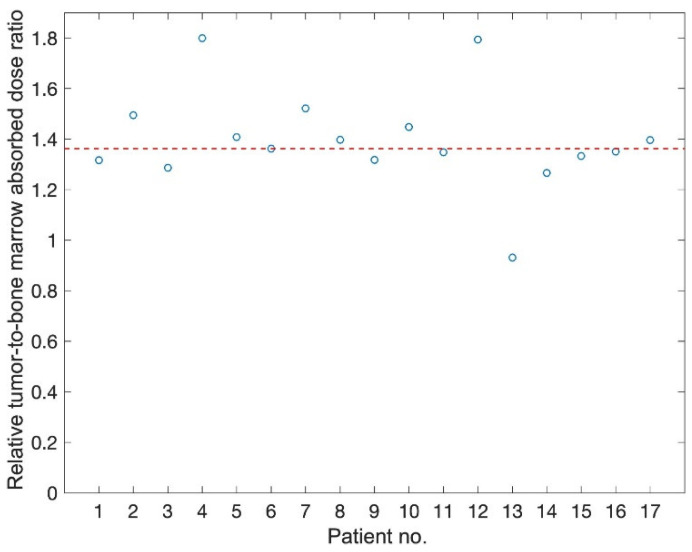
The ratio between TND_BM_ for the time interval 24 h to 28 days compared to the obtained TND_BM_ from start of treatment to infinity for the 17 patients. The median value of 136% (patient 6) is marked with a dotted red line.

**Table 1 biomedicines-09-01570-t001:** The median absorbed dose to the tumour, bone marrow and kidney within different time intervals post injection ^177^Lu-DOTATATE treatment.

Organ	Median Absorbed Dose within the Time Intervals (Range) (Gy/7.4 GBq)				
	0–inf	6 h–inf	6 h–28 d	24 h–inf	24 h–28 d
Tumour	27.3 (8.9–76.6)	26.8 (8.3–73.4)	26.2 (8.3–73.2)	24.2 (6.6–63.1)	23.6 (6.6–62.8)
		97% (93–99)	89% (96–98)	87% (75–92)	85% (75–87)
Bone marrow	0.30 (0.21–0.52)	0.23 (0.18–0.42)	0.23 (0.18–0.42)	0.18 (0.14–0.33)	0.18 (0.14–0.33)
		79% (66–95)	79% (66–95)	62% (48–80)	62% (48–80)
Kidney	4.5 (3.1–9.8)	4.2 (2.9–9.2)	4.2 (2.9–9.2)	3.5 (2.4–7.5)	4.33 (79%)
		93% (87–96)	93% (87–96)	75% (71–83)	76% (71–83)

**Table 2 biomedicines-09-01570-t002:** The median tumour-to-normal tissue absorbed dose ratio (TND) within different time intervals for treatment with ^177^Lu-DOTATATE.

Organ	TND within the Time Intervals				
	0–inf	6 h–inf	6 h–28 d	24 h–inf	24 h–28 d
Bone marrow	66.8 (42.6–296)	84.3 (41.5–410)	81.7 (41.6–408)	97.1 (39.3–461)	93.9 (39.3–459)
		122% (98–149)	121% (98–139)	140% (93–190)	134% (93–177)
Kidney	6.1 (1.3–18.7)	6.3 (1.4–19.4)	6.2 (1.4–19.3)	6.9 (1.3–21.4)	6.7 (1.3–21.3)
		105% (102–107)	103% (96–107)	115% (100–123)	112% (100–120)

**Table 3 biomedicines-09-01570-t003:** Estimated median (range) absorbed doses and biologically enhanced absorbed doses for different relative biological effectiveness (*RBE*) of the PARP inhibitor when administered within the time interval 24 h—28 days post injection of ^177^Lu-DOTATATE.

Organ	Absorbed Dose (Gy)			Enhanced Dose (Gy)		
	0–24 h	24 h–28 d	28 d–inf	*RBE* = 1	*RBE* = 1.5	*RBE* = 2
Tumour	3.1 (1.8–13.6)	23.6 (6.6–62.8)	0.3 (0.0–3.6)	27.3 (8.9–76.6)	39.1 (12.2–108)	50.9 (15.6–139)
Bone marrow	0.064	0.112	0.00 (0.00–0.00)	0.30 (0.21–0.52)	0.39 (0.29–0.69)	0.48 (0.36–0.86)
Kidney	1.17 (0.69–2.3)	3.5 (2.4–7.5)	0.00 (0.00–0.03)	4.5 (3.1–9.8)	6.2 (4.3–13.5)	8.0 (5.4–17.3)

**Table 4 biomedicines-09-01570-t004:** Estimated tumour to normal tissue enhanced dose ratio (*TNED*) for different relative biological effectiveness (*RBE*) of the PARP inhibitor when administered in the time interval 24 h–28 days post injection of ^177^Lu-DOTATATE.

Organ	Tumour-to-Normal Tissue Enhanced Dose Ratio			
	TND	TNED_2/2_ *	TNED_2/1_._5_ *	TNED_2/1_ *
Bone marrow	66.8 (42.6–296)	77.0 (41.3–348)	95.2 (53.1–423)	125 (74.3–539)
		114% (97–126)	142% (125–150)	185% (175–187)
Kidney	6.1 (1.3–18.7)	6.4 (1.3–19.8)	8.1 (1.7–25.0)	11.3 (2.3–34.0)
		105% (100–109)	134% (127–137)	185% (175–187)

* the numerator is *RBE*_tumour_ and the denominator is *RBE*_normal tissue_.

## Data Availability

The data presented in this study are available on request from the corresponding author.

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
