# Peer review of "Optimizing the Schedule of PARP Inhibitors in Combination with 177Lu-DOTATATE: A Dosimetry Rationale"

_biomedicines, 2021, doi:10.3390/biomedicines9111570_

Round 1

Reviewer 1 Report

The authors reported an optimal treatment schedule for 177Lu-DOTATATE plus PARP inhibitor combination therapy by analyzing 17 patients’ SPECT/CT and planar images to derive the absorbed doses to the tumor and normal tissues. The study concept itself is interesting, but here are a few questions/comments that I think should be addressed before it is ready for publication.

  1. The authors should provide a strong rationale that PARP inhibitors can affect the accumulation of 177Lu-DOTATATE. Otherwise, it is not reasonable to use dosimetry data to determine the optimal treatment schedule.
  2. It is well-known that SPECT/CT and planar gamma imaging have limited feasibility in providing accurate quantitative data, especially there is no ground truth (e.g. blood input function) in this study. The authors should convince us these results are suitable for further analysis.
  3. To me, the optimal treatment protocol should be determined by the treatment outcome rather than dosimetry analysis. The authors should correlate the dosimetry results with the real therapeutic efficacy to convince the reader since this is a retrospective study.

Author Response

Comments to reviewers:

Reviewer 1:

Answers/comments:

Thankyou very much for your comments.

  1. We do not propose that PARP-inhibitors would affect the accumulation of 177Lu-DOTATATE. There are preclinical data as well as theorical reasoning (e.g. p. 1 line 44-58, p. 10 line 340-348) why the effect of 177Lu-DOTATATE may be increased by adding a PARP-inhibitor like Olaparib that would decrease the ability of tumor cells to repair the 177Lu-induced DNA damage. This theory should, as you point out, be clinically tested. Our modelling study aims to assess, based on the uptake pattern of 177Lu-DOTATATE in kidney, bone marrow and tumor, how Olaparib, in theory, should be integrated with 177Lu-DOTATATE to decrease the risk of unwanted side effects from bone marrow and kidneys but still get an effect on tumor cells. The result of this modelling study is now being tested in a clinical trial which we have recently launched. We have tried to clarify that the present study is a theoretical work-up to the clinical trial (p 2. line 64-74) and p 9 line 297-298).
  2. We agree that there are uncertainties about the SPECT/CT and planar gamma imaging, but dosimetry measurements, on kidneys and tumors, with the inborn known errors, is an acknowledged method. Bone marrow dosimetry generates similar results where a correlation to bone marrow toxicity has been shown (Svensson, J. et al. EJNMMI Phys, 2016, Hagmarker, L. et al. J Nucl Med, 2019) and we think the estimates are reasonable to use for model optimization.
  3. We completely agree that the optimal treatment protocol should be determined by treatment outcome. That is why the aim of this modelling study was to launch a clinical trial assessing the combination. That phase I trial is ongoing recruiting patients but in the present manuscript the theoretical work-up on how the PARP-inhibitor, in theory, should be integrated with 177Lu-DOTATATE is accounted for. We have tried to clarify this in the manuscript (p 2. line 64-74) and p 9 line 297-298).

Reviewer 2 Report

The study by Hallqvist provides very interesting in silico data supporting a combinational therapy of PARP inhibitors with 177Lu-Dotatate for somatostatin receptor positive NEN.

I have only a few comment that I would ask the authors to reply on:

  1. Why did the authors chose to collect planar images at 2, 4, 48 and especially 168 h post injection. Was there any rational for the long time frame between 48 and 168 h?
  2. Pleas check for writing olaparib. Sometimes you are using lowercase letters and sometimes capital letters (olaparib or Olaparib).
  3. Check for typos, i.e. page 10, line 365: two full stops.

Author Response

Comments to reviewers:

Reviewer 2:

Answers/comments:

Thankyou very much for your comments.

  1. The reason for the late image (168 h) is that we want to capture the kinetics of 177-Lu-DOTATATE as best as possible, and as the half-life of 177-Lu-DOTATATE is approximately 7 days, an image at 168 h improves the estimates.
  2. and 3. Thank you for your scrutiny, we have made corrections accordingly.

Reviewer 3 Report

This paper uses radiometric modeling to propose optimum sequencing of olaparib in combination with 177Lu-DOTATATE for treatment of metastatic SRPR-positive neuroendocrine tumors.  This is desirable due to the current likelihood of relapse.  The end result is the proposed addition of the PARP inhibitor/trapping agent at 24 hours after administration of 177Lum in order to maximize impact on the tumor without introducing too much normal tissue damage, particularly on bone marrow and kidney.  This modeling outcome has already been used to propose a phase 1 clinical trial on the combination therapy.

At this point, I trust that the modeling is sufficient to optimize the therapy. The paper outlines the modeling clearly enough.  What I would like to see is a stated plan to followup as soon as possible with an assessment of the modeling based on patient data from the phase 1 trial and an indication that all  the data required to evaluate the modelling will obtained from the phase 1 trial. 

Author Response

Reviewer 3

Answer:

Thank you very much for your comment. We will follow toxicity and other clinical outcome data closely and in addition we will perform organ and tumour dosimetry to assess the trial per se and the theoretical modelling. We have added this information in the Discussion section (line 365).

Round 2

Reviewer 1 Report

The authors well respond to the issues about quantification. I understand the authors aim to answer the relationship between PARP-inhibitors and 177Lu-DOTATATE and find the optimal treatment protocol in your launched clinical trial. But, the authors should provide the launched Clinical Trial Serial Number in the manuscript to convince the readers that this “imaging-guided” dosimetry study can be useful for optimizing the treatment protocol.

Author Response

Reviewer 1:

Answer:

Thank you very much for your comment. The trial number has been added in the introduction section where the trial is mentioned (line 74). It is also mentioned in the abstract (line 33) and in the Discussion section (line 361).

Round 3

Reviewer 1 Report

Thanks for your updated information.